

# Environmental changes in Yellow River Delta with terrace construction and agricultural cropping

Dongxiao He[1,*], Jianmin Chu[2,3,*] and Hongxiao Yang[1]

[1] Qingdao Agricultural University, Qingdao, China
[2] Research Institute of Forestry, Chinese Academy of Forestry; Key Laboratory of Tree Breeding and Cultivation, National Forestry and Grassland Administration, Beijing, China
[3] Research Center for Coastal Forestry of National Forestry and Grassland Administration, Beijing, China
* These authors contributed equally to this work.

## ABSTRACT

**Background:** Crude flats in delta areas are often saline–alkaline and unsuitable for agricultural cropping. In the 1990s, people residing in the Yellow River delta constructed terraces on the flats for agricultural development. Herein, we investigated environmental changes resulting from this agricultural development and evaluated whether the current land use is effective and sustainable.

**Methods:** We sampled soil and weeds from croplands, terrace slopes, and crude flats within the delta terrace landscape. The measured soil properties included soil salinity, pH, total N, total P, and organic matter in different lands and soil layers: 0–10, 10–20, and 20–30 cm. The surveyed weed characteristics were the biomass of roots and rhizomes, species composition, life form, cover and height. These indices were statistically verified by different land types and soil layers.

**Results:** Soil salinity in the terrace crop lands was found to have reduced to $<4 \ \mathrm{g \cdot kg^{-1}}$, whereas in the crude flats, remained $>6 \ \mathrm{g \cdot kg^{-1}}$. Soil pH in the terrace croplands was surprisingly increased to $>9$ ; meanwhile, organic matter content decreased drastically, which is significantly different from that observed in the case of terrace slopes and crude flats. Total N and P content in the terrace crop lands were seemingly unchanged on averages but at the depths $>20$ cm, they reduced unavoidably. Plant underground organs were relatively scarcer in the croplands than in the other lands. Weeds grew well on the terrace slopes but were insufficient in the croplands. Overall, terrace construction is effective for developing coastal saline flats for agricultural use, but the traditional land use in the Yellow River delta has caused chronic soil degradation that is against a sustainable productive industry.

Corresponding author
Hongxiao Yang,
hongxiaoyang@126.com

## INTRODUCTION

River deltas are newly formed terrestrial lands that feed more than 500 million people worldwide with a population density of more than seven times of the global mean

(*Rahman et al., 2019*). Initially, the deltas comprise various marshes that can be used for aquaculture (*Alam, Sasaki & Datta, 2017*). With sediment depositing, they can be gradually lifted, and eventually transform into saline-alkaline lands, as observed in the Yellow River delta, China (*Yang, 2012*; *Goodbred et al., 2014*; *Liu et al., 2014*). The Yellow River delta is the largest river delta in northern China, and its size is still growing (*Fan et al., 2012*; *Mao et al., 2016*). The formerly developed areas of this delta have experienced radical changes in environmental conditions, making the aquaculture or fishery no longer viable industries. Thus, locals have to seek for alternative uses from these saline-alkaline lands.

In the formerly developed areas, the water table always appears to be underground; however, soil capillaries transport soluble minerals upward for deposition (*Sun et al., 2017*; *Moreno et al., 2018*). In the 1990s, the locals were encouraged and subsided *via* policies to build terraces with the deposited soils (*Fang, Liu & Kearney, 2005*; *Luan, 2007*; *Ning et al., 2018*). Clay was excavated from crude or original flats and was stacked as a series of raised terraces. These terraces were leveled off at heights of 2–3 m, and the remaining pits were reformed as connected ponds and ditches for collecting and draining rainfall. Concept behind this engineering is that upon raining, the rainfall can wash off the salt from terraces, thereby reducing the soil salinity of the terrace lands to facilitate crop growth (*Yang et al., 2019*).

Currently, all the terraces have been used for growing crops, such as cotton, wheat and maize for nearly 20 years, thus soil conditions must have changed (*Wang, Gong & Liu, 2004*). Soil directly provides crops with nutrients and other conditions; soil fertility is then a key indicator for evaluating environmental changes occurring in this region (*Liu et al., 2015*; *Wang et al., 2017*; *Chi et al., 2018*). Soil fertility is often dependent on properties such as salinity, pH, organic matter, nitrogen (N), phosphorous (P) and weeds (*Cierjacks et al., 2016*; *Storkey & Neve, 2018*; *Zhang et al., 2019*). Organic matter can decompose to supply crops with carbon dioxide and nutrients, and some as humus may serve as adhesive to maintain soil structure and function (*Tiessen, Cuevas & Chacon, 1994*; *Subedi, Jokela & Vogel, 2019*; *Fu et al., 2020*). However, most organic matter is ultimately supplied by weeds and crop residuals (*Stenchly et al., 2017*; *Ebabu et al., 2020*; *Jensen et al., 2020*). In addition, weeds can prevent the erosion of terrace soils because of storms.

Herein, we evaluated variations in soils and weeds within the delta terrace landscape in the early Yellow River delta. Overall, we test the hypothesis that terrace constructing may be effective in reducing soil salinity and enabling agricultural cropping. On the other hand, we further consider how to renew the land use approach for a sustainable manner. This study provides timely experiences from the Yellow River delta that are informative for worldwide people troubled by coastal saline lands.

## METHODS

### Sampling and measuring

In September 2017 and 2018, we conducted field surveys near a typical rural village (Maotou village, 118.49°E, 37.82°N) in the Yellow River delta (Fig. 1). It is approximately 80 km away from the new mouth of the Yellow River. It is in the earliest formed part of the

delta and is representative for terrace construction in this area. The annual average temperature is approximately 12 °C; the annual precipitation is approximately 580 mm and mainly occurs from June to August. The lands were made of sediments that were deposited by the old Yellow River between 1855 and 1976 (*Yang, 2012*). In the 1990s, locals started constructing terraces in this area. Currently, these terraces have been used for growing crops such as cotton and maize. These croplands are irrigated only once a year in spring for sowing, and the fresh water is drawn from the Yellow River rather than using underground water. A few areas of crude flats are still reserved for folk sacrifice ceremonies.

We selected three land types within the terrace landscape (Fig. 1): crude flats that remained in natural state; terrace slopes (30–50°), which skirt the terraces; and terrace croplands, which are leveled tops of the terraces and have been subject to tillage, cropping and fertilization for approximate 20 years. The weed survey was done in September 2017 when weeds grew quite well, and the soil survey was completed in 2018. For each land type, we surveyed weeds in 40 random quadrats (1 × 1 m), measuring the cover and height of each weed species. Then, we selected four random points from each land type for soil sampling. At each of these points, we sampled soil at three layers: 0–10, 10–20, and 20–30 cm deep. Moreover, we sampled a soil cube (10 × 10 × 10 cm) from each of these layers. These samples were taken to laboratory for further analysis. The cube samples were washed so that live roots and rhizomes, namely, underground organs (UO), remained. These UOs were oven-dried at 85 °C for 2 days and weighed using an electronic scale (precision = 0.01 g).

Other soil samples were air-dried, fully ground, and sieved with a 20-mesh wire screen. We measured pH, salinity, organic matter, total N, and total P of the sieved soil samples. We took 10 g from each sample and mixed the subsamples with pure water at the mass ratio of 1:5 (soil:water). We measured pH of steady clear solution of the subsamples using an electronic meter (ST2100/F, OHAUS in Shanghai), and measured salinity of the solution after fully drying and $H_2O_2$ solving. Organic matter (OM) in the soil was measured using the potassium bichromate titrimetric method; total N (TN) was measured using the Micro-Kjeldahl method; total P (TP) was measured using the Mo-Sb colorimetric method (*Bao, 2008*).

## Data analysis

We examined variations in the soil salinity, pH, OM, TN, TP, and dry UO among the three land types and soil layers using two-way ANOVA (*James et al., 2013*). This method requires the examined data to follow a normal distribution. As a supplement, we also conducted another analysis using the nonparametric Kruskal-Wallis test, which does not require normally distributed data (*Hollander & Wolfe, 1973*). The Kruskal-Wallis test is similar to the method of one-way ANOVA, while takes advantage of the relative ranks of the data. All these analyses were completed using the software R4.0.3 (*R Core Team, 2020*).

For evaluating the weeds, we used a composite index, the product of coverage and height (PCH) of weed species in a quadrat, thus quantifying the standard cover thickness

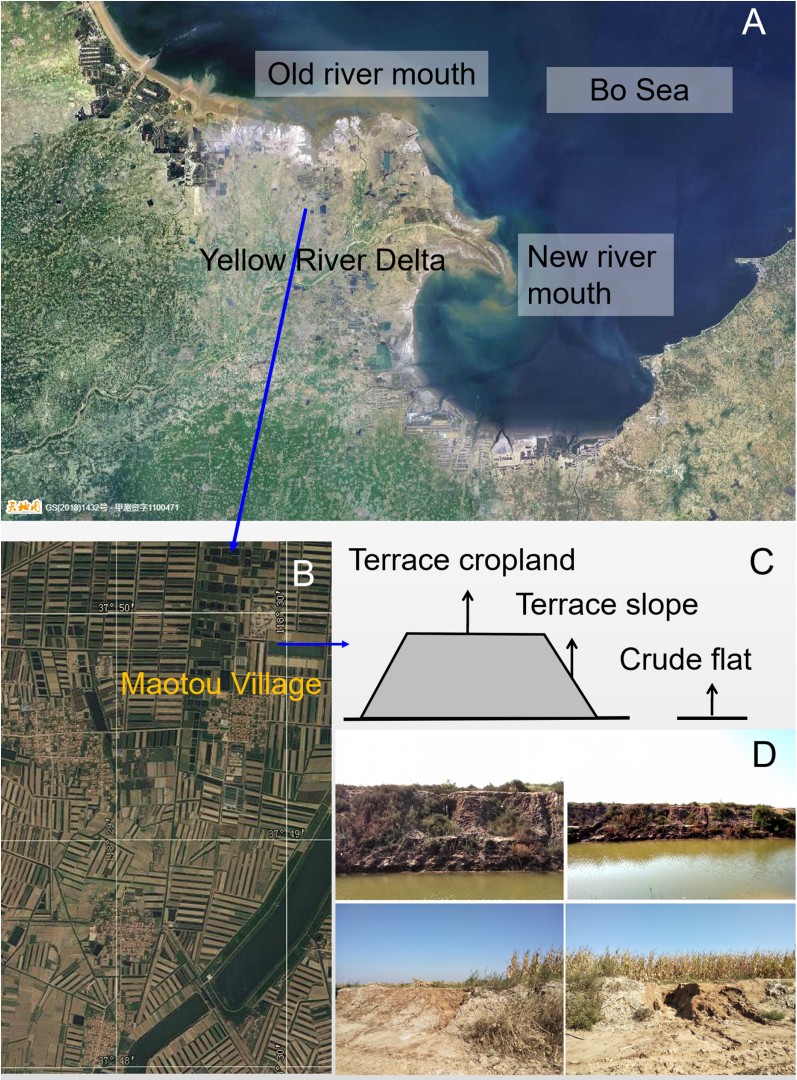

**Figure 1 Study terraces (A) in the Yellow River delta; (B) A satellite image around Maotou village (from https://www.tianditu.gov.cn, June 2021); (C) the three land types, namely, terrace cropland, terrace slope, and crude flat.** The terraces have been used for cropping for almost 20 years. The slopes are skirts of the terraces. The crude flats are reserved for sacrifice use. The right bottom photos (D) recorded some eroded slopes of the terrace croplands (taken by Hongxiao Yang).

of a weed species. Then, we classified all weeds by their life forms as follows: single (grows as a standing singleton), clump (grows as a clump of numerous tillers), vine (grows as an aboveground vine) and rhizome (grows underground with rhizomes to generate new ramets). In addition, we classified these weeds by their life span (annual *vs.* perennial). Finally, we summed the PCHs of each weed type in a quadrat and averaged the PCH sums of a certain weed type among the 40 quadrats of the same land type.

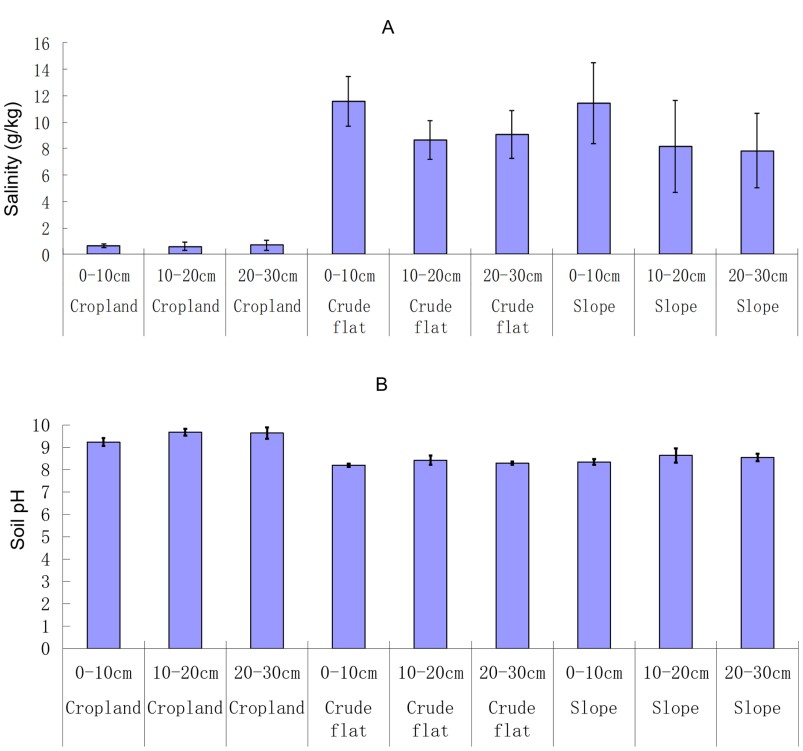

**Figure 2 Soil salinity (A) and pH (B) in different lands and soil layers.** The bar whiskers are standard errors.

## RESULTS

Soil salinity differed among the land types (ANOVA, $F_{2, 27}$ = 54.37, $p < 0.001$; Kruskal-Wallis, $x_2^2 = 23.49$, $p < 0.001$), but did not differ as significantly among the soil layers (ANOVA, $F_{2, 27} = 2.94$, $p = 0.07$; Kruskal-Wallis, $x_2^2 = 1.81$, $p = 0.41$). The interactive effect between land type and soil layer on soil salinity was not significant (ANOVA, $F_{4, 27} = 0.78$, $p = 0.55$). Soil salinity was lower in the terrace croplands (<4 g·kg$^{-1}$) than in the crude flats and terrace slopes (>6 g·kg$^{-1}$) (Fig. 2). Soil pH varied depending on land type (ANOVA, $F_{2, 27}$ = 107.54, $p < 0.001$; Kruskal-Wallis, $x_2^2 = 24.78$, $p < 0.001$) and varied almost significantly with soil layers (ANOVA, $F_{2, 27} = 6.89$, $p < 0.01$; Kruskal-Wallis, $x_2^2 = 3.48$, $p = 0.175$). The interactive effect between land type and soil layer on soil pH was not significant (ANOVA, $F_{4, 27} = 0.55$, $p = 0.70$). Soil pH in the terrace croplands was >9 and in the crude flats and terrace slopes was <9 (Fig. 2).

The total N in soil did not vary with the land type (ANOVA, $F_{2, 27} = 1.29$, $p = 0.29$; Kruskal-Wallis, $x_2^2 = 0.95$, $p = 0.623$), but varied with the soil layer (ANOVA, $F_{2, 27} = 14.64$, $p < 0.001$; Kruskal-Wallis, $x_2^2 = 17.68$, $p < 0.001$), with the highest variation in topsoil <10 cm deep (Fig. 3). The interactive effect between land type and soil layer on soil total N was not significant (ANOVA, $F_{4, 27} = 0.92$, $p = 0.47$). The N content at 20–30 cm deep was 0.23 ± 0.10 g·kg$^{-1}$ in the terrace croplands, 0.34 ± 0.03 g·kg$^{-1}$ in the crude flatlands, and 0.34 ± 0.10 g·kg$^{-1}$ in the terrace slopes. The total P content of the soil varied somewhat significantly with land type (ANOVA, $F_{2, 27} = 5.25$, $p = 0.01$;

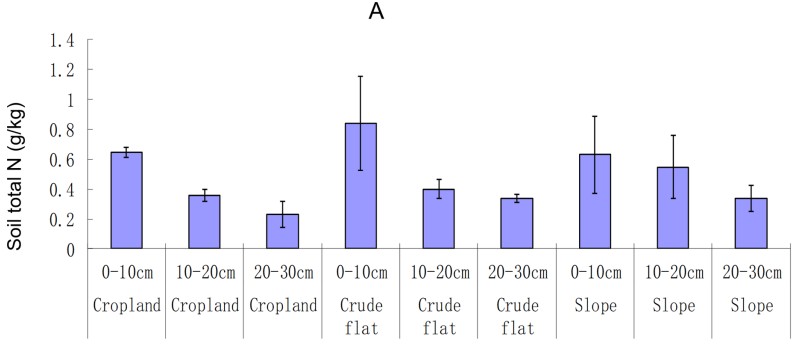

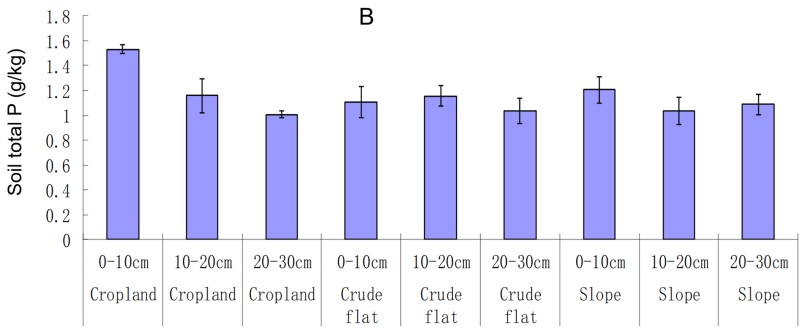

**Figure 3 Soil total N (A) and P (B) in different lands and soil layers.** The bar whiskers are standard errors.

Kruskal-Wallis, $x_2^2 = 0.77$, $p = 0.681$), and varied very significantly among different soil layers (ANOVA, $F_{2, 27} = 14.30$, $p < 0.001$; Kruskal-Wallis, $x_2^2 = 8.46$, $p = 0.015$) (Fig. 3). The interactive effect between land type and soil layer on soil total $P$ was significant (ANOVA, $F_{4, 27} = 6.50$, $p < 0.001$). The soil P content at a depth of 20–30 cm layer was $1.006 \pm 0.032$ g·kg$^{-1}$ in the terrace croplands, $1.038 \pm 0.117$ g·kg$^{-1}$ in the crude flats, and $1.087 \pm 0.097$ g·kg$^{-1}$ in the terrace slopes.

Soil OM was probably different among the land types (ANOVA, $F_{2, 27} = 3.20$, $p = 0.06$; Kruskal-Wallis, $x_2^2 = 4.95$, $p = 0.084$), and among different soil layers (ANOVA, $F_{2, 27} = 9.25$, $p < 0.001$; Kruskal-Wallis, $x_2^2 = 12.08$, $p = 0.002$). The interactive effect between land type and soil layer on soil OM was not significant (ANOVA, $F_{4, 27} = 0.81$, $p < 0.53$). The lowest OM was observed for the soil at a depth >20 cm in the terrace croplands and the highest was at topsoil (<10 cm deep) on the slopes (Fig. 4). The biomass of plant UOs varied significantly with the land type (ANOVA, $F_{2, 27} = 9.28$, $p < 0.001$; Kruskal-Wallis, $x_2^2 = 12.84$, $p = 0.002$) and varied nearly significantly with the soil layer (ANOVA, $F_{2, 27} = 8.54$, $p = 0.001$; Kruskal-Wallis, $x_2^2 = 5.52$, $p = 0.063$). The interactive effect between land type and soil layer on plant UOs was significant (ANOVA, $F_{4, 27} = 3.76$, $p = 0.01$). The biomass of plant UOs was extremely low in the terrace croplands (Fig. 4).

The weed types mainly comprised perennials and rhizomes (Fig. 5). The relative cover thickness of weeds was the greatest on the terrace slopes (>600 cm), and the lowest in the terrace croplands (<50 cm). Weeds did not grow well (<100 cm) in the crude flats; they

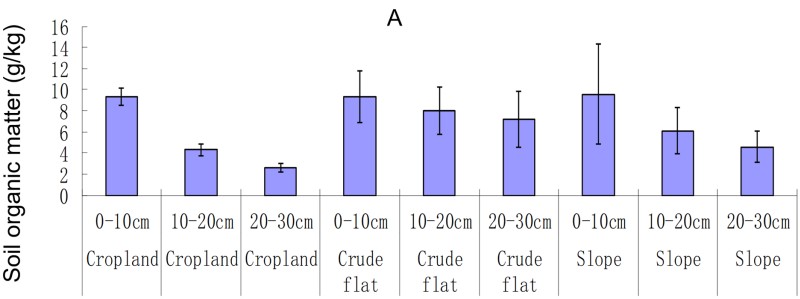

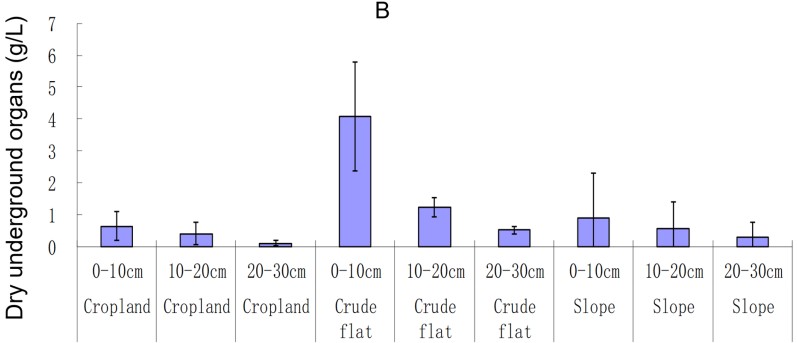

**Figure 4 Soil organic matter (A) and dry plant undergroundorgans (B) in different lands and soil layers.** Dry plant underground organs were measured by the biomass (g) of dry plant (weed) roots and rhizomes in the soil (L). The bar whiskers are standard errors.

were only slightly better than in the terrace croplands. The richness of weed species showed a similar pattern, with 23 species found on the slopes, 16 species found in the terrace croplands, and 10 species found in the crude flats.

## DISCUSSION

### Positive changes with the terracing and cropping

Soil salinity in the terrace croplands was significantly lower than in the crude flats. After terraces were built using the deposited soils, rainfall must have accelerated the desalination process. Some years later, salinity was no longer a hindrance to growing certain varieties of cotton and maize; thus, the locals began growing them on the terraces. Harvesting the crops could also contribute to salinity reduction in the lands (*Zhao et al., 2013*). Presently, soil salinity in the terrace croplands has reduced to <4 g·kg$^{-1}$, whereas in the crude flats, remains >6 g·kg$^{-1}$.

Weeds on the terrace slopes grew considerably thicker than in the flats and croplands, and the richness of weed species was also the highest on the terrace slopes. This is partly attributed to farmers' intention on permitting weed growth because weeds (particularly perennials) can protect the terraces from potential eroding and collapsing for storms (*Liu et al., 2013*; *Asimeh et al., 2020*). In the croplands, weeds were scarce because of farmers' tillage and weeding activities; in the crude flats, they grew only slightly better than

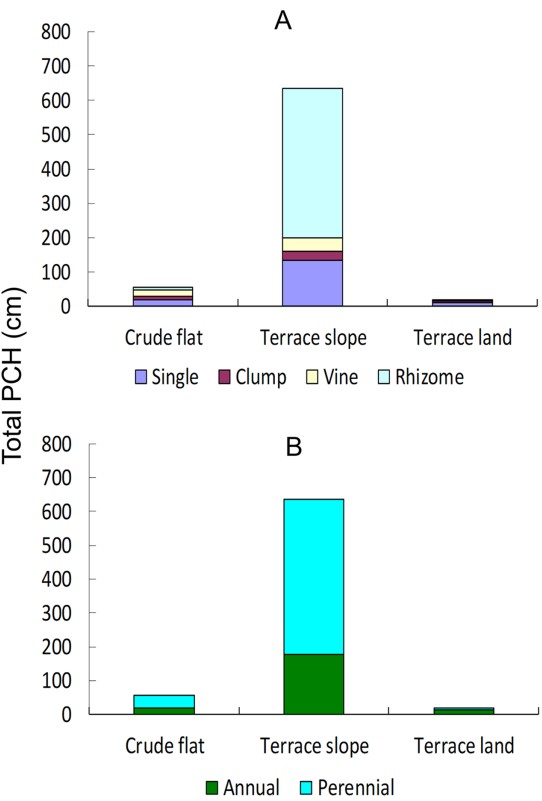

**Figure 5 Cumulative PCH of weeds in different lands and soil layers.** PCH, the product of coverage and height of weed species. Total PCH, total of PCHs of all species in a plot. (A) In different life forms; (B) in different life spans.

in the croplands perhaps owing to high soil salinity and compaction (*Bennett, Barrett-Lennard & Colmer, 2009*; *Xia et al., 2009*). This indicates that terrace slopes can be refuges for conserving a considerable number of plant species, which can protect the croplands against rainfall erosion.

## Negative changes with the farming

Unfortunately, soil pH in the terrace croplands was higher (>9) than in the crude flats (<9). Soil OM content, particularly at depths >10 cm, was lower in the croplands than in the flats. These changes must reduce soil fertility in the croplands (*Chapagain & Raizada, 2017*; *Zhang et al., 2020*). They are considered to be a chronic response to the current farming regime, rather than to the terracing practice, because these changes were not as evident in the terrace slopes.

Because of agricultural practices in the terrace croplands, fertilizers and herbicides were repeatedly added to eradicate weeds and to replenish N and P contents in the soils. However, these chemicals could cause cumulative changes in the soils (*Zalidis et al., 2002*; *Kaur, Kaur & Chauhan, 2018*). Some of the commonly used chemical fertilizers, such as diammonium phosphate and urea, can slightly increase soil pH (*Zhou et al., 2004*; *Jin et al., 2008*). The increase may be small initially; however, it could accumulate year after year, and now has turned so significant. Aboveground and underground amounts of

weeds (plants) were observed to be extremely low in the croplands, thus the supply of soil organic matter from them must be not enough to compensate for the yearly decomposition of soil OM that must have been hastened by the tillage (*Hobley et al., 2018*; *Storkey & Neve, 2018*).

### Obscure changes of soil total N and P

The total N and P contents in the soils seemed to be equivalent among the three land types. As the most-required elements by crops, soil N and P must be repeatedly extracted from the soils with annual crop harvest. To compensate for the loss of N and P, locals added chemical fertilizers, such as diammonium phosphate and urea. The N and P contents in the soil seemed to be maintained and were not significantly lower than in the crude flats and terrace slopes. However, this is only a judgment based on averages.

The N and P contents in the soil were dependent on soil depths. For soil layers at the depth of 20–30 cm, N and P were observed to be much lower in the croplands than in other lands. This indicates that the added chemical fertilizers could not supplement the N and P loss in the subsoil >20 cm deep. If the loss cannot be timely reversed, soil degradation will become more and more severe, and ultimately make the terrace lands unsuitable for agriculture production (*Dai et al., 2017*; *Jensen et al., 2020*). As thus, we think that it is time to convert the current land use regime into a new one that should well restore fertility of the terrace soils (*Ordóñez-Fernández et al., 2018*; *Fu et al., 2020*). For example, locals can be encouraged to grow legume crops or forage such as alfalfa whose roots can feed deep soil with multifunctional biotic fertilizers, avoiding excessively relying on chemical fertilizers (*Hu et al., 2019*). This suggestion is also informative for other coastal areas annoyed with thus saline lands.

## CONCLUSION

After the delta saline soil was stacked as raised terraces, soil salinity decreased steadily, and with the terracing, abundant weeds began to grow on the new habitats of slopes. However, long-term agricultural practices in the terrace lands have caused profound degradation of the lands, such as increased soil pH, soil organic matter loss, and unavoidable loss of total N and P in subsoil (>20 cm).

## ACKNOWLEDGEMENTS

We gratefully acknowledge Liwen Bianji for editing the article.

### Funding

This study was funded by Qingdao Agricultural University and Fundamental Research Funds of CAF (CAFYBB2020SZ001-3). The funders had no role in study design, data collection and analysis, decision to publish, or preparation of the manuscript.

## Grant Disclosures

The following grant information was disclosed by the authors:
Qingdao Agricultural University and Fundamental Research: CAFYBB2020SZ001-3.

## Competing Interests

The authors declare that they have no competing interests.

## Author Contributions

- Dongxiao He performed the experiments, analyzed the data, prepared figures and/or tables, and approved the final draft.
- Jianmin Chu performed the experiments, prepared figures and/or tables, and approved the final draft.
- Hongxiao Yang conceived and designed the experiments, analyzed the data, authored or reviewed drafts of the paper, and approved the final draft.

## Data Availability

The raw measurements are available in the Supplemental File.

## Supplemental Information

Supplemental information for this article can be found online at http://dx.doi.org/10.7717/peerj.12469#supplemental-information.

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
