# Peer review of "Environmental changes in Yellow River Delta with terrace construction and agricultural cropping"

_PeerJ, doi:10.7717/peerj.12469_

## Round 0.1 · original submission · Major Revisions

Dear Dr. Yang and co-authors,

I just received reviews of your manuscript. Although both reviewers consider the study very interesting and significant, some issues need to be considered before acceptance. Please, consider all comments and suggestions provided by both reviewers during the revision of your manuscript.

A comprehensive revision of the English of the manuscript is necessary before submitting the new version.

Don't forget to include a letter response along with the revised version of the manuscript. In this letter you must respond point by point to each question.

Best regards,

Xiaoming Kang

·

Basic reporting

The article is fluent in English and conforms to the structure of an academic paper.

Experimental design

Research question have a certain value, but the experimental design is not systematic.
For example, the experimental design did not consider the differences in crops, and there is no long-term data support;why did the author choose to do the weed survey in September instead of other representative months?

Validity of the findings

The conclusion is not well stated, and the content of the statement has little to do with the research results.
L.188: There are no related results of soil erosion to support this part.
L.190: The result of soil nutrients in the article is 0-30cm, not greater than 20cm.
L.190: There are no indicators to support the sustainability of the three land use types, and the author does not propose how to improve soil fertility.

Additional comments

Figure 1 does not clearly show the three types of land use. It is recommended to add photos or explain more clearly.
The author also needs to explain or answer several key questions:
1. Whether the input of irrigation, chemical or organic fertilizers etc. can make the local agriculture sustainable development, there is no deep discussion in the article;
2. The Yellow River Delta has a large area. What is the area and representativeness of the land use types discussed in this article?
3. Can the research content in the article support the title and conclusion?

Reviewer 2 ·

Basic reporting

This study investigated the environmental changes result from the terrace agriculture in Yellow River delta in China. The result showed that the soil salinity has been efficiently reduced by the terrace construction. Weeds on the terrace slopes can protect the terraces with roots and rhizome. Overall, the result is interesting and provides useful information for sustainable agriculture. This paper is well organized with technical text structure and figures, but some details in describing the methods and results need to be improved.

Experimental design

The experiment overall was well designed and conducted rigorously, but some details need to complement.
1.LINE 58: “sustainability of landuse” was proposed to be evaluated, but there was no relative index on the sustainability found in this paper. Therefore, I suggested reword this sentence and remove the “sustainability evaluation”.
2.LINE 60-61: hypothesis for this study was supposed. However, the sentence ”however.......” is not a hypothesis, but a conclusion. I suggest remove this sentence. Moreover, the word “useful” in line 60 is ambiguous. Please suppose specific points (e.g. reduce salinity, improve the soil nutrients) instead of “useful”.
3.LINE 67: The field survey was conducted in September 2017 and 2018. If the field survey had been conducted twice, the year effect was not presented in the results. Author need give more details in processing the data from two years.
4.LINE 89: the manufacturer and producing place of instruments should be given (here for ST2100).
5.LINE 94-95: I suggested the soil layer can be treated as fixed factor, rather than a covariate, because the soil layers for each sample are the same and are always considered as a main factor in explaining the variation of soil properties. Thus, two-way ANOVA can be applied for testing the effect of land type and soil layers, as well as their interactions, on soil properties.

Validity of the findings

The result were explicit and conclusions were made based on the data, here’s only one point on the Figures organization.
1.Figure 2-4: Since the land type was treated as a main factor in explaining the changes in soil properties, the soil properties of different land type and soil layers can be presented in one unstacked bar chart (land type as X axis and soil layer as groups). Maybe this way is easier for readers to compare the difference among land types and soil layers.

Additional comments

1.The title was unclear, and the key word “time” was even not studied in this research. If author wish to emphasize the “timely revolution”, the chronic changes of cropland ecosystem need to be studies. Here, I suggest use a simple but explicit sentence such as “the changes in ....on terrace cropland”

2.Author discussed the positive, negative and obscure changes in soil properties and weeds of the terrace cropland comparing with the crude flat and slope land. However, only the decrease of salinity on the terrace cropland and strong growth of weeds on slope land can be explained by the terrace construction. The increase of pH, decrease of soil organic matter and changes of soil N and P were all attributed to the fertilization and tillage, rather than the terrace construction. Therefore, I suggest the discussion can be reorganized as 1. effect of terrace construction, 2. effect of tillage. This structure can be more fit to the aim of this study.

3.Author mentioned “weeds were insufficient to compensate for the decomposition of soil organic matters in the cropland”. However, I can not find this point neither in result nor discussion. Please give more explanation on this point in the discussion.

Annotated reviews are not available for download in order to protect the identity of reviewers who chose to remain anonymous.

---

## Round 0.2 · Minor Revisions

Dear Dr. Yang and co-authors,

I just received the reviews of your manuscript. Please, consider all comments and suggestions provided by Reviewer 1 during the revision of your manuscript. More reasonable explanation of the experimental design is also needed before the acceptance.

Don't forget to include a letter response along with the revised version of the manuscript. In this letter you must respond point by point to each question.

Best regards,

Xiaoming Kang

·

Basic reporting

no comment

Experimental design

The experimental design has major flaws.
“ In September, the crops are harvested but weeds are left. Before it, the crops is high and immature, the farmers do not allow the investigation” is not a good explanation.

Validity of the findings

no comment

Additional comments

Figure 1 still does not clearly show the three types of land use. It is recommended to add Field photos.
Fig. 4 What are dry plant underground organs? Need to explain in full text.
"The surveyed weed characteristics were the biomass of roots and rhizomes, species composition, life form, cover and height" is Not fully demonstrated and explained in the results and discussion section.

Reviewer 2 ·

Basic reporting

This paper has been greatly improved from the first version, and all the comments have been solved.

Experimental design

no comment

Validity of the findings

no comment

Additional comments

no comment

---

## Round 0.3 · Minor Revisions

Dear authors,

Thank you for your resubmission to PeerJ.

Jasmine Janes, the Section Editor, has commented and said for your revised manuscript:

"The figures (2-5) state that they are total values but they show some type of error bar, which suggests that they are not a total value but a mean.

Also, the authors will need to clarify what the whiskers are on the bar plots - are they SE, SD or CI, etc. The y-axis labels should be moved to the y-axis, they should not be at the top of the figure like a heading would be. Each plot on each figure will need an A and B designation to discriminate between them (e.g., A is pH while B is salinity), and these designations, plus the description of what the whiskers are, need to be reflected in the figure legends. Figure 5 needs the plot headings removed. Also, in Figure 2, one plot has whiskers and one doesn't, why?

Lastly, the introduction does a really nice job of stating that the data from this manuscript will be helpful in combatting/understanding salinity impacts in coastal freshwaters around the world. However, this is not brought up again in the discussion. This topic should be included in the discussion and highlighted - how exactly are these data helpful for people in saline lands? Are there particular approaches that can be recommended (e.g., is the application of fertilizer actually a positive or negative? - that isn't 100% clear at the moment. What is meant by "convert the current land use to a new one"? - does that mean stop agricultural activity altogether? These points could be more clearly conveyed to improve and widen the prospective readership of the manuscript."

Please, consider all comments and suggestions provided by the section editor during the revision of your manuscript.

Don't forget to include a letter response along with the revised version of the manuscript. In this letter you must respond point by point to each question.

Best regards

Xiaoming Kang

---

## Round 0.4 · accepted · Accept

Dear authors,

I am pleased to inform you that, following the revision made based on the reviewer’s comments, your manuscript is now acceptable for publication in PeerJ.

Best regards

Xiaoming Kang